# Isolation of ESBL-Producing *Enterobacteriaceae* in Food of Animal and Plant Origin: Genomic Analysis and Implications for Food Safety

**DOI:** 10.3390/microorganisms13081770

**Published:** 2025-07-29

**Authors:** Rosa Fraccalvieri, Stefano Castellana, Angelica Bianco, Laura Maria Difato, Loredana Capozzi, Laura Del Sambro, Adelia Donatiello, Domenico Pugliese, Maria Tempesta, Antonio Parisi, Marta Caruso

**Affiliations:** 1Istituto Zooprofilattico Sperimentale della Puglia e della Basilicata, via Manfredonia 20, 71121 Foggia, Italy; rosa.fraccalvieri@izspb.it (R.F.); stefano.castellana@izspb.it (S.C.); maria.difato@izspb.it (L.M.D.); loredana.capozzi@izspb.it (L.C.); laura.delsambro@izspb.it (L.D.S.); adelia.donatiello@izspb.it (A.D.); domenico.pugliese@izspb.it (D.P.); antonio.parisi@izspb.it (A.P.); marta.caruso@izspb.it (M.C.); 2Department of Veterinary Medicine, University Aldo Moro of Bari, Strada per Casamassima Km 3, Valenzano, 70010 Bari, Italy; maria.tempesta@uniba.it

**Keywords:** food, Extended Spectrum β-Lactamase (ESBL), ESBL-Producing *Enterobacteriaceae* (ESBL-PE), antimicrobial resistance (AMR)

## Abstract

**Background**: The spread of ESBL-producing *Enterobacteriaceae* (ESBL-PE) strains in food poses a potential risk to human health. The aim of the study was to determine the occurrence of ESBL-PE and to investigate their distribution on foods. **Methods:** A total of 1000 food samples, including both raw and ready-to-eat products, was analyzed for the presence of ESBL-producing *Enterobacteriaceae* using chromogenic selective agar. Antibiotic resistance in the isolated strains was assessed using conventional methods, while whole-genome sequencing was employed to predict antimicrobial resistance and virulence genes. **Results**: The overall occurrence of ESBL-PE strains was 2.8%, with the highest contamination in raw meat samples (10%). A total of 31 multidrug-resistant (MDR) strains was isolated, mainly *Escherichia coli*, followed by *Klebsiella pneumoniae*, *Salmonella enterica*, and *Enterobacter hormaechei*. All strains exhibited high levels of resistance to at least four different β-lactam antibiotics, as well as to other antimicrobial classes including sulfonamides, tetracyclines, aminoglycosides, and quinolones. Whole-genome sequencing identified 63 antimicrobial resistance genes, with *bla*_CTX-M_ being the most prevalent ESBL gene. Twenty-eight (90%) isolates carried Inc plasmids, known vectors of multiple antimicrobial resistance genes, including those associated with ESBLs. Furthermore, several virulence genes were identified. **Conclusions**: The contamination of food with ESBL-PE represents a potential public health risk, underscoring the importance of the implementation of genomic surveillance to monitor and control the spread of antimicrobial resistance.

## 1. Introduction

The production of enzymes capable of hydrolyzing and inactivating β-lactam antibiotics is the most important mechanism of resistance in the *Enterobacteriaceae* family. Extended-spectrum β-lactamases (ESBLs) are enzymes that confer resistance to penicillins, first- to third-generation cephalosporins and monobactams (e.g., aztreonam), but not to cephamycin (e.g., cefoxitin) or carbapenems (e.g., meropenem, imipenem) [1]. However, these enzymes are inhibited by β-lactamase inhibitors such as clavulanic acid [2,3]. The first ESBLs were identified in the 1980s as variants of the TEM-1 and SHV-1 enzymes, resulting from specific amino acid substitutions in *K. pneumoniae* and *E. coli* strains, respectively [4,5,6]. In recent years, novel ESBLs have emerged, with the cefotaximase-M (CTX-M) family becoming the most prevalent group in many European countries. Among these, CTX-M-15 is the most frequently detected variant. Over 140 TEM and 60 SHV derivatives capable of hydrolyzing third-generation cephalosporins and aztreonam have been identified [5,7]. CTX-M-type ESBLs are now the most widely distributed globally, particularly in *K. pneumoniae* and *E. coli*, which are major pathogens in both healthcare-associated and community-acquired infections [8]. These enzymes are not exclusively chromosomally encoded but are frequently carried on mobile genetic elements, especially plasmids. Notably, *bla*_CTX-M_-like genes are often co-located with other antimicrobial resistance determinants, such as *aac (6’)-Ib-cr*, *bla*_OXA_, *catB*, *tet*, *aadA*, *dfrA17*, *sul* genes [9]. ESBL-producing *Enterobacteriaceae* are often multidrug-resistant, with additional resistance to non-β-lactam antibiotics such as aminoglycosides (e.g., amikacin, gentamicin, streptomycin), fluoroquinolones (e.g., ciprofloxacin), trimethoprim, tetracyclines, sulfonamides (e.g., sulfisoxazole), and chloramphenicol [10,11]. The spread of ESBL-producing *Enterobacteriaceae* in food represents a potential threat to human health, as the resistance genes are frequently associated with mobile genetic elements capable of transferring to both commensal and pathogenic bacteria, even across different genera and species [12].

This study aims to enhance our understanding of the occurrence of ESBL-producing *Enterobacteriaceae* across various food categories by screening samples collected from the southern Italian regions of Apulia and Basilicata, and to assess differences in contamination levels between raw and ready-to-eat foods. Antibiotic resistance in the isolated strains will be assessed using conventional phenotypic methods, while whole-genome sequencing (WGS) will be employed to predict the presence of resistance genes. Comparing results from both approaches will aid in the classification of isolates based on their resistance profiles and contribute to a deeper understanding of antimicrobial resistance in the food chain.

## 2. Materials and Methods

### 2.1. Sampling

Samples were collected using a non-probabilistic convenience sampling method. All samples were transported in temperature-controlled containers maintained at 4 ± 2 °C and promptly delivered to the laboratory. A total of 1000 food samples was analyzed, consisting of 500 raw and 500 ready-to-eat (RTE) products. All samples were sourced from the retail market across the southern Italian regions of Apulia and Basilicata between 2018 and 2023.

### 2.2. Bacterial Strain Isolation

The food samples were analyzed with the aim of detecting ESBL-producing *Enterobacteriaceae*, as described in a previous study [13] with the exception that CHROMagar™ ESBL (CHROMagar, Paris, France) was used as the chromogenic isolation medium.

### 2.3. ESBL-Producing Screening

All presumptive ESBL-producing *Enterobacteriaceae* strains, previously isolated, were evaluated for ESBL production by determining the Minimum Inhibitory Concentration (MIC), in accordance with the Clinical and Laboratory Standards Institute (CLSI) guidelines [14]. The Sensititre™ EUVSEC^®^ panel (Trek Diagnostic Systems, Westlake, OH, USA), which includes the ESBL-producing screening combination test method, was employed. A reduction of two or more dilution steps in the MIC of cefotaxime and/or ceftazidime in the presence of clavulanic acid, compared to the MIC of cephalosporin alone, was interpreted as indicative of ESBL production.

### 2.4. Whole-Genome Sequencing

For each sample, DNA was purified using the QIAmp DNA mini kit (Qiagen, Hilden, Germany), according to the manufacturer’s instructions. DNA quality was assessed by calculating the absorbance ratio at 260/280 nm, while DNA quantification was performed by measuring the optical density (OD) at 260 nm using a BioPhotometer (Eppendorf, Milan, Italy). DNA samples purified with an A260/A280 ratio between 1.8 and 2.0 were considered to be of high quality and stored at −20 °C until analysis. Libraries were prepared using the Illumina DNA Prep kit (Illumina, San Diego, CA, USA) and sequenced using a MiSeq platform (Illumina, San Diego, CA, USA) with a 2 × 250 bp paired end approach [15]. Bioinformatic analyses were conducted through the European Galaxy server (https://usegalaxy.eu, accessed on 18 March 2025). Raw reads were assembled using SPAdes version 3.15 [16] and assembly quality was assessed with Quast version 5.0.2 [17]. Genome assemblies were subsequently analyzed to screen for (*i*) ribosomal multilocus sequence typing (rMLST) by PubMLST species identification database (https://pubmlst.org/species-id, accessed on 21 March 2024), (*ii*) multilocus sequence typing (MLST) [18], and (*iii*) detection of antimicrobial resistance (AMR) genes and point mutations, virulence genes (VGs) and plasmid sequences using both ABRIcate (Galaxy Version 1.0.1) and StarAMR (Galaxy Version 0.11.0) tools. Default parameters were applied for all tools.

The nucleotide sequences of strains were deposited in GenBank under the BioProject accession PRJNA1076372 (Appendix A).

### 2.5. Antimicrobial Susceptibility Test

The ESBL-producing strains were analyzed for their resistance profiles against the antibiotics listed in Table 1, by the minimum inhibitory concentration (MIC) method, following the CLSI guidelines for Enterobacterales as reported in a previous study [13]. *E. coli* ATCC 25,922 was used as the control strain. MIC values were interpreted according to the guidelines provided by the CLSI [14].

### 2.6. Statistical Analysis

Differences in the percentages of isolates were compared using the Chi-square test, performed with Epi Info software version 3.3.2. A *p*-value < 0.05 was considered statistically significant.

## 3. Results

### 3.1. Contamination Rate of Food Samples with ESBL-Producing Enterobacteriaceae

A total of 1000 food samples was cultured to isolate ESBL-producing *Enterobacteriaceae* (ESBL-PE). Strains were successfully recovered from 5% (25/500) of the raw food samples and from 0.6% (3/500) of the ready-to-eat (RTE) food samples (Table 2 and Table 3). A statistically significant difference was observed between these two groups (χ^2^ = 17.8; *p* < 0.05). Although all samples were cultured for the detection of ESBL-producing *Enterobacteriaceae*, no strains were isolated from raw vegetables, RTE dried or cooked sausages, RTE ready meals or RTE bakery and pastry products. The highest contamination rate was observed in raw meat samples (10%), in which the occurrence of ESBL-PE was significantly higher compared to other food categories (*p* < 0.05). A total of 31 ESBL-PE strains were isolated from 28 (2.8%; 28/1000) food samples. Specifically, 27 strains were isolated from 25 raw food samples, with two samples yielding two strains each. Four strains were isolated from 3 RTE food samples, including one ice cream sample, from which two strains were isolated (Table 2 and Table 3).

Draft assemblies of the isolated strains, combined with rMLST analysis of the draft genomes, enabled accurate taxonomic assignment (Table 2 and Table 3). *Escherichia coli* (20/31; 65%) was the most commonly isolated species, followed by six *Klebsiella pneumoniae* (6/31; 19%), two *Enterobacter hormaechei* (2/31; 6%), two *Salmonella enterica* (2/31; 6%) and one *Klebsiella quasipneumoniae* (1/31; 3%). MLST analysis identified 16 sequence types (STs) among *E. coli* strains, five STs in *K. pneumoniae* and one ST each in *E. hormaechei* and *S. enterica* strains. For one *K. quasipneumoniae* strain, the sequence type could not be determined (Table 2 and Table 3).

### 3.2. Antimicrobial Resistance Profiles of the ESBL-Producing Enterobacteriaceae Isolates

Antimicrobial susceptibility testing (including 31 antibiotics grouped in 11 classes) was conducted on 31 strains and the results are collected in Figure 1. All (100%) strains showed a multidrug resistance (MDR).

The two *E. hormaechei* isolates exhibited intrinsic resistance to ampicillin (AMP), amoxicillin/clavulanic acid (AUG2), ampicillin/sulbactam (A/S2), cefazolin (FAZ), and cefoxitin (FOX). Seven *K. pneumoniae* isolates showed intrinsic resistance to ampicillin. The highest resistance rates were observed for ampicillin, cefazolin and cefotaxime, with all 31 isolates (100%) resistant. This was followed by resistance to piperacillin (*n* = 30; 97%), ceftriaxone and sulfisoxazole (*n* = 28; 90%), tetracycline (*n* = 27; 87%) and both ampicillin/sulbactam and aztreonam (*n* = 24; 77%). For fifteen additional antimicrobials, resistance rates ranged from 71% to 13% (Figure 2). No resistance was observed to piperacillin/tazobactam, carbapenems, amikacin or tigecycline.

### 3.3. Antimicrobial Resistance Determinants

This investigation identified 63 antimicrobial resistance genes (ARGs) using the AMR ResFinder database in combination with the ABRIcate tool. These ARGs were further grouped into 14 classes based on their antibiotic resistance profiles (Figure 3). All isolates carried at least one gene encoding ESBL β-lactamases (*bla*_CTX-M_, *bla*_SHV_*, bla*_TEM_). Additionally, ARGs conferring resistance to aminoglycosides, phenicols, macrolides, quinolones, sulfonamides, and tetracyclines were detected in all isolates (Figure 3). A strong correlation was observed between genotypic predictions and phenotypic resistance profiles. Specifically, there was 100% concordance for β-lactamase genes, while 27 out of 28 sulfonamide-resistant isolates and 25 out of 27 tetracycline-resistant isolates harbored genes associated with phenotypical resistance. Additionally, mutations in *gyrA* (S83L; D87N), *parC* (A56T; S80I; S80R) and *parE* (L416F; S458A) were detected in *E. coli* strains (Table 4). Among these, the non-synonymous mutation S83L in *gyrA* was identified in eight *E. coli* strains, with three of these also carrying the D87N mutation. The S80I substitution in *parC* was the most prevalent. The two substitutions in *parE* were identified in two strains of *E. coli*.

### 3.4. Correlation Between the Presence of ARGs and Phenotypic Resistance

All ESBL-producing isolates carried extended-spectrum β-lactamase (ESBL) genes (*bla*_CTX-M_; *bla*_SHV_; *bla*_TEM_). Among the 31 isolates, 21 (66%) harbored the *bla*_CTX-M_ gene. The most prevalent were those encoding enzymes from the CTX-M-1 group, particularly CTX-M-15 and CTX-M-1, found in 17 isolates (55%), with CTX-M-15 alone detected in 12 isolates (39%). Genes from the CTX-M-9 group (CTX-M-14, CTX-M-55, CTX-M-65) and the CTX-M-8 group were less frequent, with frequencies of 10% (3/31) and 3% (1/31), respectively. Thirteen isolates (42%) carried ESBL *bla*_SHV_ genes, while the ESBL variant *bla*_TEM-106_ gene was detected in only one isolate (3%). Additionally, genes encoding the non-ESBL enzyme TEM-1 were present in 14 isolates (45%). Two *E. hormaechei* strains displayed phenotypic resistance to first-generation cephalosporins and cefoxitin, consistent with the presence of the *bla*_ACT_ gene encoding AmpC β-lactamases.

Further correlations between genotype and phenotype were observed for other antibiotic classes. Aminoglycoside resistance genes (*aac*, *aph*, *aad*, *ant*) were identified in 28 isolates (90%), of which 22 exhibited resistance to streptomycin, tobramycin, or gentamicin. Chloramphenicol resistance genes (*catA*, *catB*, *clmA*) were found in 12 isolates (39%), with 11 also exhibiting phenotypic resistance. However, two *E. coli* strains were phenotypically resistant despite lacking any known chloramphenicol resistance genes. Sulfonamide resistance genes (*dfrA*, *floR*, *sul*) were detected in 27 isolates (87%), all of which were phenotypically resistant to sulfisoxazole and/or trimethoprim/sulfamethoxazole. One *E. coli* isolate, however, showed phenotypic resistance to sulfisoxazole without any corresponding ARGs. Tetracycline resistance genes (*tet*) were identified in 25 strains (81%), according with phenotypic resistance. Notably, two *E. coli* strains exhibited resistance to tetracycline in the absence of identifiable *tet* genes. Quinolone resistance genes (*qnr*) were present in 13 isolates (42%), of which 11 demonstrated phenotypic resistance to ciprofloxacin, levofloxacin, or nalidixic acid. Conversely, eight *E. coli* and two *S. enterica* isolates were phenotypically resistant despite lacking the *qnr* gene, while two isolates carried the *qnr* gene but remained phenotypically susceptible. Additionally, non-synonymous mutations were identified in 4% of *E. coli* strains, suggesting a role in resistance to fluoroquinolones. Macrolide resistance genes (*mef*, *mph*) were detected in eight isolates (26%), with phenotypic resistance observed in four. Two strains showed phenotypic resistance to azithromycin despite the absence of corresponding resistance genes. The *fosA* gene, conferring resistance to fosfomycin, was identified in nine isolates (29%). However, phenotypic resistance to fosfomycin was not assessed in this study. Finally, 29 isolates (94%), excluding the two *S. enterica* strains, harbored efflux pump genes (*oqxA*, *oqxB*, *mdfA*), which are known to contribute to multidrug resistance.

### 3.5. Detection of Plasmid Genes

Plasmid gene prediction was performed using the PlasmidFinder database. It is important to note that only partial plasmid nucleotide sequences were detected in the analyzed strains, rather than complete plasmid assemblies. A total of 105 plasmid replicons or replicon fragments were identified across the isolates (Figure 4). The most frequently detected replicon was ColRNAI_1, present in 14 strains, followed by IncFIB (AP001918)_1, IncI1_1_Alpha and IncFII, found in 12, 11 and 7 strains, respectively. Notably, strain ESBL101 harbored the highest number (*n* = 7) of distinct replicon types. In contrast, no replicon sequences were detected in two *E. hormaechei* strains (ESBL134 and ESBL139) and one *E. coli* strain (ESBL087) (Figure 4).

### 3.6. Detection of Virulence Genes

The virulence genes (VGs) prediction was carried out using the Virulence Factors Database (VFDB). A total of 191 VGs, encoding adhesins, siderophores/iron transport systems, toxins, secretion systems, invasins or others, was identified in the isolates (Appendix A and Figure 5). Regardless of taxonomic species, the outer membrane protein gene *omp*A proved the most prevalent, detected in 100% of the isolates, followed by 29 (94%) isolates that carried *ent* and *fep* genes necessary for the biosynthesis and transport of the siderophore enterobactin; 22 (71%) strains carried *csg* genes and 23 (74%) isolates showed *yag/ecp* genes, both essential for surface adhesion, biofilm formation and interaction with host cells. The distribution of the number of VGs per strain is reported in Figure 5. Notably, the two *S. enterica* strains exhibited the highest number of virulence genes (117), indicating a potentially enhanced pathogenicity. These strains were distinguished by the presence of genes encoding adhesins (e.g., *fim*, *ipf*, *csg*, *fae*), siderophores and iron transport systems (e.g., *ent*, *fep*, *irp*, *ybt*, *fyu*), invasins (*mgt*, *mig*), as well as a variety of toxins and secretion systems.

Among the *E. coli* isolates, the number of virulence genes ranged from 78 to 37. Most *E. coli* strains harbored genes encoding adhesins (e.g., *fim*, *cgs*, *yag*/*ecp*, *fde*), siderophores/iron transport systems (e.g., *ent*, *fep*, *fes*), secretion systems (e.g., *gsp*), and invasins systems (e.g., *esp*). *K. pneumoniae* strains exhibited a lower level of virulence genes, ranging from 21 to 10 genes. All the *K. pneumoniae* strains carried *yag*V/*ecp*A-E for adhesins genes, *ent* and *fep* for siderophores/iron transport systems and *ykg*K/*ecp*R for other virulence genes. The two *E. hormaechei* strains carried only a single virulence gene: the adhesins gene *omp*A.

## 4. Discussion

This study investigated the occurrence of extended-spectrum β-lactamase-producing *Enterobacteriaceae* (ESBL-PE) in 1000 food samples purchased in the Southern Italian regions of Apulia and Basilicata between 2018 and 2023. Half of the samples (n = 500) were ready-to-eat (RTE) foods, with a low positivity rate (0.3%), while the remaining 500 were raw foods, among which 5% tested positive. The overall prevalence of ESBL-PE contamination was 2.8% (28/1000). Notably, raw meat samples showed a significantly higher positivity rate (10%) compared to other food categories (range: 0.0–4.0%; *p* < 0.05). Among the 207 raw meat samples, poultry meat (n = 50) showed the highest contamination rate, with 32% (16/50) testing positive. This result is consistent with international reports, which have documented poultry meat contamination rates of 36% in Ghana, 21% in Brazil and the United States, and 25.9% in Switzerland [19,20]. Even higher prevalence rates have been reported in parts of Europe, reaching 93% in Spain and 80% in the Netherlands [21,22].

A statistically significant decreasing trend in the prevalence of ESBL-producing bacteria in food-producing animals has been reported by the European Food Safety Authority and the European Centre for Disease Prevention and Control (EFSA-ECDC) with a notable decline in the prevalence of ESBL-producing *E. coli* in Italy, from 82.3% during 2017–2018 to 43.4% in the 2021–2022 period [23].

In our study, among RTE foods, ESBL-PE were detected in one cheese sample, one salad sample and one ice cream sample. No positive samples were identified in other RTE foods investigated. Among raw foods, 4% of milk samples tested positive, while no ESBL-PE were detected in raw vegetables. A recent study reported contamination rates of about 18.0% in raw milk, 1.3% in cheese and 0.25% in vegetables [24]. Notably, the raw milk contamination was relatively low in Europe (2%), Malaysia (3.18%), and India (6.6%) [25,26,27].

Among the food samples that tested positive for ESBL-PE, a total of 31 bacterial isolates were obtained. Whole-genome sequencing revealed four species: *E. coli*, predominantly isolated from raw meat, with the highest frequency in poultry meat; *K. pneumoniae* and *K. quasipneumoniae*, primarily detected in turkey and pork meat; *S. enterica*, exclusively isolated from poultry meat; and *E. hormaechei*, isolated from ice cream and raw seafood.

In a similar study, *E. coli* accounted for 61% of isolates, while *K. pneumoniae* represented 9.1% [20]. A study conducted in the United Kingdom reported a prevalence of ESBL-producing *E. coli* in 27.5% of analyzed meat samples [28]. In Germany, the highest prevalence of ESBL-producing *E. coli* was observed in chicken meat (74.9%), followed by turkey meat (40.1%) [24]. The prevalence of ESBL-producing *K. pneumoniae* in animal-derived food products shows substantial variation, with reported rates ranging from 3% in retail chicken meat samples in the Netherlands and 3.7% in chicken liver samples in Algeria, to 23.4% in raw milk samples in Lebanon and 25% in food fish samples in India [29,30,31,32]. The ability of *Salmonella* spp. to produce ESBLs, similar to other members of the *Enterobacteriaceae* family, has led the World Health Organization (WHO) to classify them as high-priority pathogens [33]. Two studies published in 2022, one from Poland and one from Italy, reported relatively high prevalence rates (14.3% and 11.7%, respectively) of ESBL-producing *Salmonella* spp. in poultry meat [34,35]. In contrast, lower prevalence rates have been reported in studies from Canada, China, Egypt and Southeast Asia, with contamination levels in foods of animal origin ranging from 0.05% to 3.5% [1,36,37,38].

In this study, *Enterobacter hormaechei*, a member of the *Enterobacter cloacae* complex (ECC), was the least frequently isolated species. According to the literature, ECC strains are known to produce extended-spectrum β-lactamases (ESBLs) and represent the third most prevalent group of drug-resistant Enterobacterales, following *Escherichia coli* and *Klebsiella pneumoniae* [39].

In this study, MLST analysis of 31 isolates identified distinct sequence types (STs), reflecting a high degree of genetic diversity. However, in four isolates the STs could not be determined due to unmatched allelic profiles in the Achtman MLST database. The most frequently identified STs among *E. coli* were ST10 (*n* = 3), ST69 (*n* = 2), and ST1011 (*n* = 2), while the remaining STs were represented by single isolates. Among the STs identified in *E. coli*, ST10 represents a globally disseminated lineage commonly associated with both commensal and extraintestinal pathogenic strains, often harboring extended-spectrum β-lactamase (ESBL) genes [40,41]. Notably, an *E. coli* ST10 isolate was recovered from an RTE cheese sample. ST69 is a well-known uropathogenic clone, frequently associated with antimicrobial resistance [42], while ST155 has been implicated in zoonotic transmission and multidrug resistance [43,44]. In *K. pneumoniae*, the detection of ST29 is of particular concern, as this lineage has been implicated in nosocomial outbreaks and is known to carry carbapenemase-producing plasmids [45]. Both *S. enterica* isolates belonged to ST32, corresponding to serovar Infantis, a major foodborne pathogen known for its association with multidrug resistance [46]. The presence of these high-risk clones underscores the importance of continuous genomic surveillance to monitor the spread of clinically significant lineages.

All strains isolated in this study were tested for antimicrobial susceptibility against 31 antibiotics. As expected for ESBL-producing isolates, the highest resistance levels were observed among β-lactam antibiotics. High resistance rates were also observed among non-β-lactam antibiotics, including sulfonamides, tetracycline, aminoglycosides and fluoroquinolones. The production of extended-spectrum β-lactamases (ESBLs) is frequently associated with co-resistance to other antibiotic classes, such as aminoglycosides (e.g., amikacin, gentamicin, streptomycin), fluoroquinolones (e.g., ciprofloxacin), trimethoprim, tetracyclines (e.g., tetracycline), sulfonamides (e.g., sulfisoxazole), and chloramphenicol [11]. All isolates were classified as multidrug-resistant (MDR), since they exhibited resistance to at least one antibiotic in three or more distinct antimicrobial classes [47].

Additionally, genomic analysis identified 63 ARGs potentially associated with phenotypic resistance. The most frequently detected genes were those conferring resistance to aminoglycosides (*aac*, *aph*, *aad*, *ant*) and β-lactams (*bla_ACT_*; *bla*_CTX-M_; *bla*_SHV_; *bla*_TEM_), followed by genes associated with resistance to sulfonamides (*dfrA*, *floR*, *sul*), efflux pumps (*oqxA*, *oqxB*, *mdfA*), tetracyclines (*tet*), phenicols (*catA*, *catB*, *clmA*), fosfomycin (*fosA*), macrolides (*mef*, *mph*) and quinolones (*qnr*). Among this class of antibiotics, non-synonymous mutations in *gyrA*, *parC* and *parE* were predicted, but only in some *E. coli* strains. The co-occurrence of the S83L and D87N substitutions in *gyrA* was detected in three *E. coli* isolates, while the S83L mutation alone was identified in five additional isolates. This substitution has also been reported as the most common in other studies [48,49]. Both variants are associated with fluoroquinolone resistance and evidence suggests that the mutation at amino-acid position 83 confers a higher level of resistance than the one at position 87 [50,51]. Nevertheless, the co-occurrence of mutations in *gyrA* or in combination with other genes may contribute to higher levels of resistance. In six *E. coli* isolates, substitutions were also identified in *parC* and *parE*, two genes mapped within the quinolone resistance-determining regions (QRDRs), suggesting their involvement in fluoroquinolone resistance [52].

When comparing phenotypic antimicrobial resistance with genomic sequencing-based predictions, a notable degree of concordance was observed. All isolated strains harbored at least one ESBL gene (*bla*_CTX-M_; *bla*_SHV_; *bla*_TEM_). The CTX-M gene was detected in 66% of the isolates, with CTX-M-15 being the most common variant, identified in 39% of the strains. Similar findings have been reported in previous studies, where CTX-M-type ESBLs were predominant in *E. coli, K. pneumoniae* and *Salmonella enterica* serovar Typhimurium [53,54]. In European countries, ESBL production in clinical isolates of human origin is most frequently mediated by CTX-M-15 enzymes, whereas their presence remains relatively low in isolates from livestock [55,56]. A study conducted in the UK reported the absence of CTX-M-15-producing *E. coli* in food samples derived from both animal and non-animal sources [28]. In another study, the *bla*_CTX-M-15_ gene was detected in 5.2% (*n* = 21) of all strains isolated from foods [9]. SHV-type ESBLs remain among the most commonly detected β-lactamases in Enterobacterales, particularly in *K. pneumoniae* and *E. coli*, while TEM-type ESBLs have largely declined in detection rates [5].

A strong correlation between the presence of ARGs and phenotypic resistance was particularly evident for β-lactamases, sulfonamides, aminoglycosides and tetracyclines. However, the presence of acquired ARGs in bacterial genomes does not always correlate directly with phenotypic resistance. The detection of amino acid mutations associated with fluoroquinolone resistance may explain the phenotypic resistance observed in strains where multiple mutations co-occurred. Conversely, phenotypic resistance may be observed in the absence of detectable ARGs. This discrepancy may be attributed to other resistance mechanisms not assessed in this study [57,58].

Genomic sequencing revealed a total of 191 virulence genes among the isolated strains, associated with various mechanisms of virulence that contribute to their ability to cause infections. These genes encode proteins involved in key virulence functions, such as adhesion to surfaces, evasion of the host immune response, cellular invasion and other processes that ultimately contribute to disease development [59].

Consistent with previous studies, all isolates harbored the *ompA* gene, which encodes an outer membrane protein involved in key processes such as biofilm formation, adhesion and invasion of eukaryotic cells, modulation of host immune responses and contribution to antibiotic resistance [60]. All *S. enterica* and *E. coli* strains tested were positive for the main virulence genes involved in the synthesis of adhesin proteins, such as intimin, fimbriae and curli which are critical for attachment to host cells during colonization. Additionally, these strains possessed genes encoding iron acquisition systems, particularly enterobactin, that facilitate iron uptake from the host, a mechanism frequently employed by Enterobacterales [61,62].

The two *S. enterica* strains also carried invasion-associated genes encoding proteins involved in cell invasion, allowing them to enter and survive within host cells. Furthermore, they contained multiple genes associated with secretion systems and toxins, which are known to contribute to severe diarrheal diseases [63].

Among the *E. coli* strains, the only detected toxin gene was *ast*A, which encodes the enteroaggregative heat-stable toxin 1 (EAST1), a toxin implicated in diarrheal illness [64]. Notably, one *E. coli* strain carrying this gene was isolated from an ice cream sample. All seven *K. pneumoniae/K quasipneumoniae* isolates were positive for adhesin and iron acquisition system genes but tested negative for toxin and secretion system genes. In this study, a total of 33 different plasmid replicons or replicon fragments were identified among the analyzed strains. Notably, 90% of the isolates harbored Inc plasmids, which are known to carry multiple antimicrobial resistance (AMR) genes, including those encoding extended-spectrum β-lactamases (ESBLs). IncFII plasmids, which frequently carry the *bla*_CTX-M-15_ gene and are known for their high transferability, were detected in two *E. coli* and five *K. pneumoniae* strains isolated from raw meat samples [65,66].

## 5. Conclusions

The findings of this study highlight the presence, albeit relatively low overall, of extended-spectrum β-lactamase-producing *Enterobacteriaceae* (ESBL-PE) in food products intended for human consumption, with a particularly concerning occurrence in raw poultry meat (32%). This result, consistent with other reports, indicates that such products may represent a relevant source of human exposure to multidrug-resistant (MDR) bacteria. The widespread detection of MDR strains exhibiting high levels of resistance to β-lactams and other commonly used antimicrobial classes raises serious concerns regarding food safety and the future effectiveness of antimicrobial therapies.

The associated risk extends beyond foodborne infections to include silent colonization of the human gastrointestinal tract, facilitating horizontal gene transfer of antimicrobial resistance genes (ARGs) to both commensal and pathogenic bacteria, contributing to the broader spread and persistence of resistance within the human population. Of particular concern is the identification of the *bla_CTX-M-15_* gene, frequently associated with highly transferable IncFII plasmids and widely reported in clinical isolates across Europe; its detection in *E. coli* and *K. pneumoniae* strains from raw meat reinforces the hypothesis of a direct connection between the food chain and public health.

Moreover, the co-occurrence of multiple virulence genes (e.g., *ompA*, siderophores, adhesins), although the gene expression has not been evaluated, suggests that these isolates are not only resistance carriers but also possess considerable pathogenic potential, particularly threatening to immunocompromised individuals. These findings suggest that food products may act as vehicles for the transmission of MDR *Enterobacteriaceae* capable of causing difficult-to-treat infections. The identification of MDR phenotypes, mobile resistance determinants such as IncFII plasmids and clinically significant ESBL genes (e.g., *bla_CTX-M-15_*, *bla_SHV_*, *bla_TEM_*), jointly with virulence factors, underscores the urgent need for a comprehensive One Health approach that integrates human, animal and environmental health.

The global spread of ESBL-producing *Escherichia coli* has been recognized as a critical public health threat by the World Health Organization (WHO) due to its rapid dissemination and major clinical implications [33,67]. In this context, our data could add valuable genomic information to the global effort to monitor the emergence and transmission of ESBL-producing *Enterobacteriaceae* in Southern Italy and may serve as a reference for comparative analyses of this One Health-associated clone. In this regard, the application of next-generation sequencing and in silico genome analysis using global sequence typing and source-tracking databases is crucial for elucidating the origins, evolution and spread of clinically relevant high-risk clones and their resistance mechanisms.

## Figures and Tables

**Figure 1 microorganisms-13-01770-f001:**
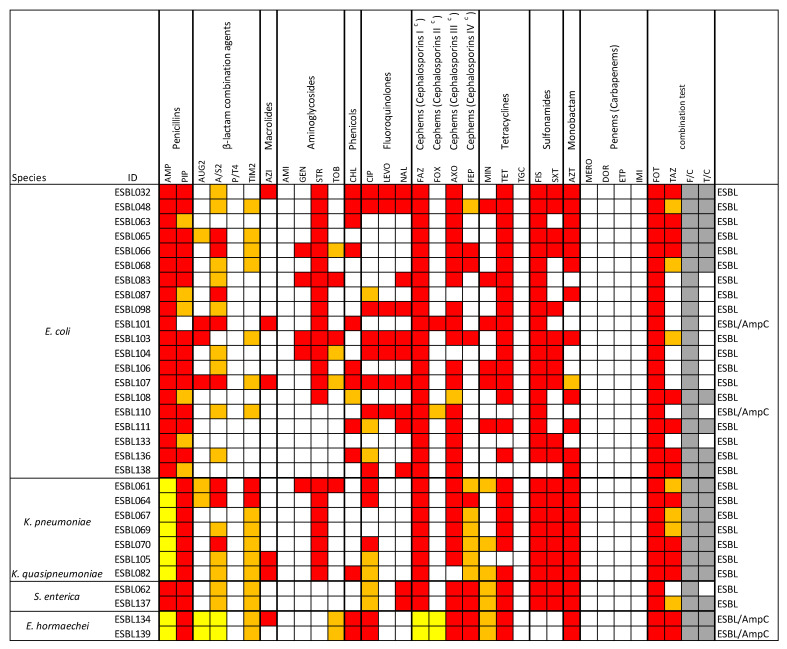
Antimicrobial resistance revealed on 31 strains of *Enterobacteriaceae*. The species are indicated in the first column. Antimicrobial agents, grouped by drug class, are listed in the remaining columns. c = subclass of generation I, II, III, or IV. Red: Resistance; yellow: intrinsic resistance; white: susceptibility; orange: intermediate resistance; gray: positive result in the combination test for ESBL production. AMP: Ampicillin; PIP: Piperacillin; AUG2: Amoxicillin/Clavulanic acid; A/S2: Ampicillin/Sulbactam; P/T4: Piperacillin/Tazobactam; TIM2: Ticarcillin/Clavulanic acid; AZI: Azithromycin; AMI: Amikacin; GEN: Gentamicin; STR: Streptomycin; TOB: Tobramycin; CHL: Chloramphenicol; CIP: Ciprofloxacin; LEVO: Levofloxacin; NAL: Nalidixic Acid; FAZ: Cefazolin; FOX: Cefoxitin; AXO: Ceftriaxone; FEP: Cefepime; MIN: Minocycline; TET: Tetracycline; TGC: Tigecycline; FIS: Sulfisoxazole; SXT: Trimethoprim/Sulfamethoxazole; AZT: Aztreonam; MERO: Meropenem; DOR: Doripenem; ETP: Ertapenem; IMI: Imipenem; FOT: Cefotaxime; TAZ: Ceftazidime; F/C: Cefotaxime/Clavulanic acid; F/C: Ceftazidime/Clavulanic acid.

**Figure 2 microorganisms-13-01770-f002:**
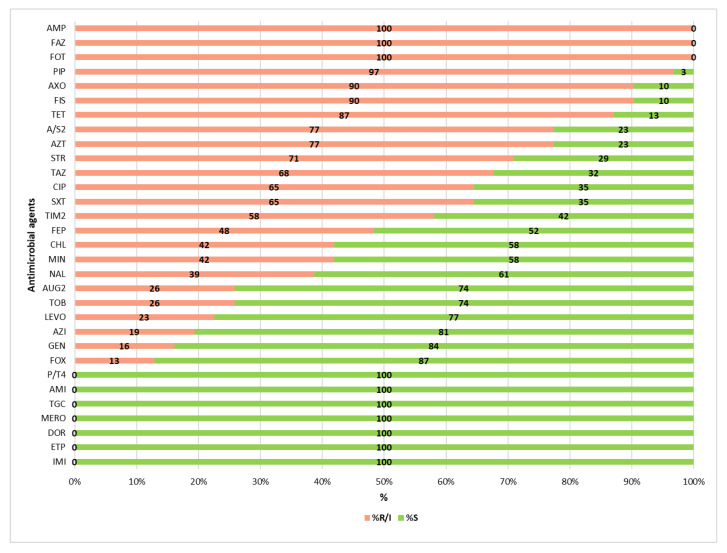
Antimicrobial resistance distribution. R = resistant; I = intermediate-resistant; S = susceptible. Tested antibiotics: AMP: Ampicillin; FAZ: Cefazolin; FOT: Cefotaxime; PIP: Piperacillin; AXO: Ceftriaxone; FIS: Sulfisoxazole; TET: Tetracycline; A/S2: Ampicillin/Sulbactam; AZT: Aztreonam; STR: Streptomycin; TAZ: Ceftazidime; CIP: Ciprofloxacin; SXT: Trimethoprim/Sulfamethoxazole; TIM2: Ticarcillin/Clavulanic acid; FEP: Cefepime; CHL: Chloramphenicol; MIN: Minocycline; NAL: Nalidixic Acid; AUG2: Amoxicillin/Clavulanic acid; TOB: Tobramycin; LEVO: Levofloxacin; AZI: Azithromycin; GEN: Gentamicin; FOX: Cefoxitin; P/T4: Piperacillin/Tazobactam; AMI: Amikacin; TGC: Tigecycline; MERO: Meropenem; DOR: Doripenem; ETP: Ertapenem; IMI: Imipenem.

**Figure 3 microorganisms-13-01770-f003:**
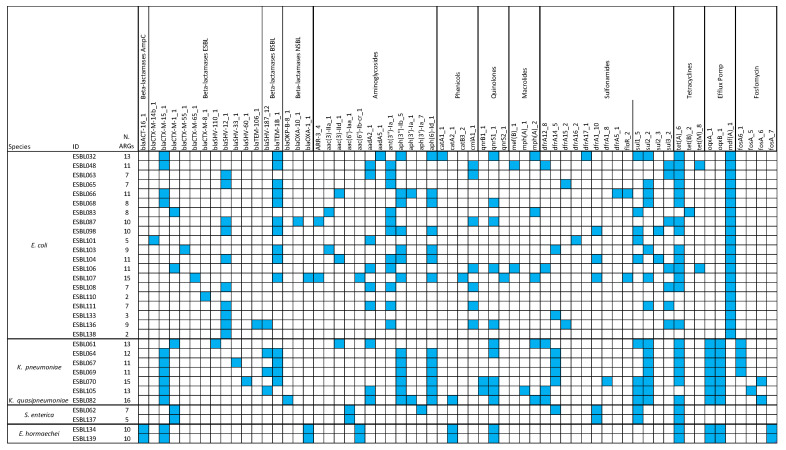
Representation of resistome profile related to the 31 strains (ESBL#ID). The first column lists the species predicted by WGS. Additionally, the table reports the AMR genes identified by WGS. Strains that harbored AMR genes, identified by WGS, are highlighted in blue.

**Figure 4 microorganisms-13-01770-f004:**
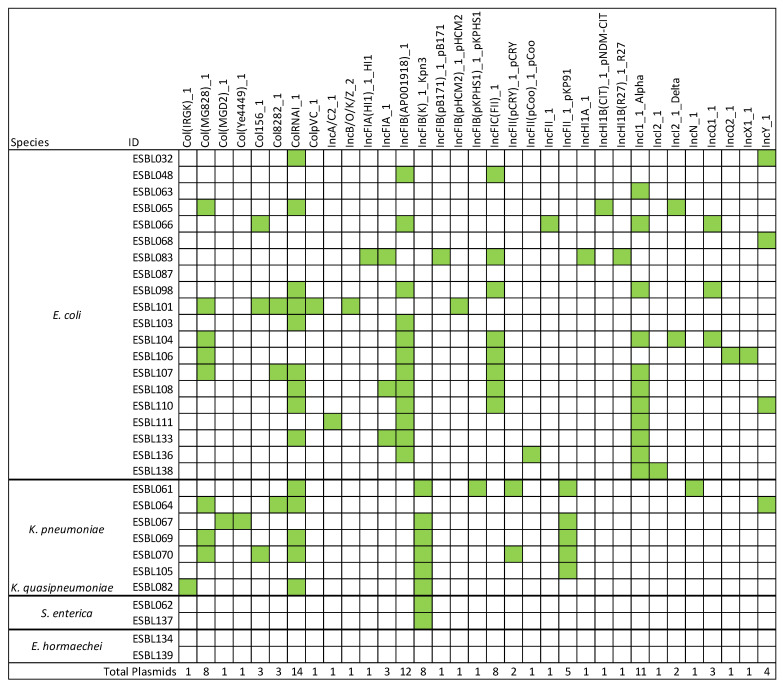
Schematic view of replicons related to the 31 strains (ESBL#ID). For each strain, the presence of a replicon is shown in green.

**Figure 5 microorganisms-13-01770-f005:**
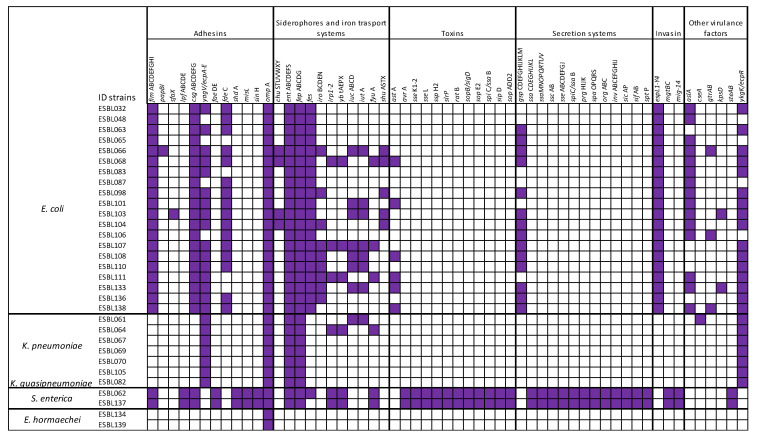
Schematic view of virulence genes related to the 31 strains (ESBL#ID). For each strain, the presence of VGs is shown in purple.

**Table 1 microorganisms-13-01770-t001:** Antimicrobials tested.

Antimicrobial Class (Subclass)	Agent	Range µg/mL
Penicillins	AMP—Ampicillin	1–32
	PIP—Piperacillin	16–64
β-lactam combination agents	AUG2—Amoxicillin/Clavulanic acid	1/0.5–32/16
	A/S2—Ampicillin/Sulbactam	4/2–16/8
	P/T4—Piperacillin/Tazobactam	8/4–128/4
	TIM2—Ticarcillin/Clavulanic acid	8/2–64/2
Macrolides	AZI—Azithromycin	0.25–32
Aminoglycosides	AMI—Amikacin	2–8
	GEN—Gentamicin	0.25–16
	STR—Streptomycin	2–64
	TOB—Tobramycin	2–8
Phenicols	CHL—Chloramphenicol	2–32
Quinolones (Fluoroquinolones)	CIP—Ciprofloxacin	0.015–4
	LEVO—Levofloxacin	1–8
Quinolones	NAL—Nalidixic Acid	0.5–32
Cephems (Cephalosporins I ^c^)	FAZ—Cefazolin	1–16
Cephems (Cephalosporins II ^c^)	FOX—Cefoxitin	0.5–64
Cephems (Cephalosporins III ^c^)	AXO—Ceftriaxone	0.25–64
Cephems (Cephalosporins IV ^c^)	FEP—Cefepim	0.06–32
Tetracyclines	MIN—Minocycline	1–8
	TET—Tetracycline	4–32
Tetracycline (Glycylcycline)	TGC—Tigecyline	1–8
Sulfonamides	FIS—Sulfisoxazole	16–256
	SXT—Trimethoprim/Sulfamethoxazole	0.12/2.38–4/76
Monobactam	AZT—Aztreonam	1–16
Penems (Carbapenems)	MERO—Meropenem	0.03–16
	DOR—Doripenem	4–8
	ETP—Ertapenem	0.25–8
	IMI—Imipenem	0.12–16
ESBL-producing screening combination test method	FOT—Cefotaxime	0.25–64
	TAZ—Ceftazidime	0.25–128
	F/C—Cefotaxime/Clavulanic acid	0.06/4–64/4
	T/C—Ceftazidime/Clavulanic acid	0.12/4–128/4

**Table 2 microorganisms-13-01770-t002:** ESBL-producing *Enterobacteriaceae* isolated from raw food samples. N. = number. ID = Identification. * One *E. coli* and one *Salmonella enterica* were revealed in the same poultry meat sample; ** one *E. coli* and one *K. pneumoniae* were isolated from the same minced turkey sample.

Raw Food Samples				Isolates			
Source	N. Analyzed Samples	N. ESBL Positive (Type of Food)	ESBL Positive (%)	N. ESBL Positive	rMLST Species	Sequence Type (ST)	ID Strain
**Raw Milk**	**52**	2 (raw milk from vending machine)	4	2	*E. coli*	ST744	ESBL032
					*E. coli*	ST10	ESBL048
**Raw Meat**	**207**	10 (poultry meat)		9	*E. coli*	ST223	ESBL063
					*E. coli*	ST1011	ESBL098
					*E. coli*	ST457	ESBL103
					*E. coli*	ST1011	ESBL104
					*E. coli*	ST155	ESBL108
					*E. coli*	ST88	ESBL110
					*E. coli*	ST8132	ESBL111
					*E. coli*	ST93	ESBL133
					*E. coli* ***	ST4162	ESBL136
				2	*S. enterica* ***	ST32	ESBL137
					*S. enterica*	ST32	ESBL062
		2 (turkey meat)		2	*K. pneumoniae*	ST3069	ESBL070
					*K. pneumoniae*	ST187	ESBL105
		1 (minced turkey meat)		2	*E. coli* ****	ST2705	ESBL065
					*K. pneumoniae* ****	ST29	ESBL064
		1 (minced horse meat)		1	*E. coli*	ST69	ESBL068
		2 (minced beef and pork)		2	*E. coli*	ST69	ESBL066
					*K. pneumoniae*	ST35	ESBL067
		1 (pork meat)		1	*K. pneumoniae*	ST29	ESBL069
		2 (fresh pork sausage)		2	*K. pneumoniae*	ST25	ESBL061
					*E. coli*	ST2179	ESBL107
		1 (hamburger)		1	*E. coli*	ST218	ESBL106
**Subtotal**		20	10	22			
**Seafood products**	**133**	1 (mussel)		2	*E. coli*	ST10	ESBL101
		1 (frozen squid rings)			*E. hormaechei*	ST109	ESBL134
**Subtotal**		2	1.5	2			
**Bakery and pastry products, fresh pasta**	**28**	1 (fresh egg pasta)	3.6	1	*E. coli*	ST515	ESBL087
**Vegetables**	**80**	0	0	0	/		/
**TOT**	**500**	**25**	**5**	**27**			

**Table 3 microorganisms-13-01770-t003:** ESBL-producing *Enterobacteriaceae* isolated from RTE food samples. N. = number. ID = Identification. ND = Not Detected. One *E. coli* and one *E. hormaechei* were isolated from the same packaged ice cream.

Ready-To-Eat Food Samples				Isolates			
Source	N. Analyzed Samples	N. ESBL Positive (Type of Food)	ESBL Positive (%)	N. ESBL Positive	rMLST Species	Sequence Type (ST)	ID Strain
**Milk and Cheese**	**240**	1 (canestrello cheese)	0.5	1	*E. coli*	ST10	ESBL083
**Dried or cooked Sausages**	**35**	0	0	0	/		/
**Ready meals**	**100**	0	0	0	/		/
**Bakery and pastry products, fresh pasta**	**27**	0	0	0	/		/
**Ice cream**	**66**	1 (packaged ice cream)	1.5	2	*E. coli*	ST2223	ESBL138
					*E. hormaechei*	ST109	ESBL139
**Vegetables**	**32**	1 (mixed salad)	3.0	1	*K. quasipneumoniae*	ND	ESBL082
**TOT**	**500**	**3**	**0.6**	**4**			

**Table 4 microorganisms-13-01770-t004:** Mutations detected in the isolates. The presence of each mutation is marked in red. Green color indicates the absence of mutations.

	*gyr A*		*parC*			*parE*	
Isolate ID	S83L	D87N	A56T	S80I	S80R	L416F	S458A
ESBL032							
ESBL048							
ESBL061							
ESBL062							
ESBL063							
ESBL064							
ESBL065							
ESBL066							
ESBL067							
ESBL068							
ESBL069							
ESBL070							
ESBL083							
ESBL087							
ESBL098							
ESBL101							
ESBL103							
ESBL104							
ESBL105							
ESBL106							
ESBL107							
ESBL108							
ESBL110							
ESBL111							
ESBL133							
ESBL134							
ESBL136							
ESBL137							
ESBL138							
ESBL139							
ESBL82							

## Data Availability

The data that support the findings of this study are available from the corresponding author upon reasonable request.

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
