# Peer review of "Isolation of ESBL-Producing Enterobacteriaceae in Food of Animal and Plant Origin: Genomic Analysis and Implications for Food Safety"

_microorganisms, 2025, doi:10.3390/microorganisms13081770_

Round 1

Reviewer 1 Report

Comments and Suggestions for Authors

Thank you for the opportunity to review this manuscript. The study addresses an interesting topic related to food safety and antimicrobial resistance. However, before it can be considered for publication, several issues must be addressed. 

First, the authors state that the aim of the study is to determine the prevalence of ESBL-producing Enterobacteriaceae. However, prevalence implies the use of a representative and adequately sized population sample, which does not appear to be the case here. The term frequency would be more appropriate. Additionally, it should be clearly stated in the Materials and Methods that a convenience sampling strategy was used. I suggest including a dedicated subsection titled "Sampling" before the section "Isolation of bacterial strains."

Second, β-lactams encompass several antibiotic classes, including penicillins, cephalosporins, monobactams, and carbapenems. Thus, the statement that all strains were resistant to β-lactams is misleading, as resistance to some β-lactams does not imply resistance to all. Please revise this point for clarity and accuracy.

One minor issue... The sentence “This resistance is mediated by a limited number of mutated resistance genes that encode one or more β-lactamases...” is unclear. Are the authors referring to naturally occurring genes or mutated variants? Please clarify what is meant by "mutated" in this context.

I believe that the rationale behind the study is not sufficiently developed. It currently reads as an exploratory investigation, which weakens the impact. The authors should better articulate the relevance of the study and its implications for public health and food safety to justify its publication. Also, there are several similar studies already published. Why this should be approved? 

Third, the WGS methodology section lacks critical information. The sequencing platform and the bioinformatics tools used should be clearly stated, even if the analysis is detailed in a cited reference (10.3390/microorganisms13010163). A minimum description of these methods is essential for transparency and reproducibility. Also, consider merging this section with "2.5 Nucleotide Sequence Accession Numbers" for better structure.

The rationale for testing multiple antibiotics from the same class is unclear. According to CLSI guidelines (cited by the authors), a single representative is typically sufficient for each class. For example, testing four carbapenems appears redundant unless justified. Please explain why multiple agents from the same class were used.

Line 134: The following sentence needs to be revised for clarity “The most frequently isolated species was Escherichia coli (20/31; 65%), followed by Klebsiella pneumoniae (7/31; 23%), Enterobacter hormaechei (2/31; 6%), and Salmonella enterica (2/31; 6%).”

I could not find any results or discussion regarding potential mutations in gyrA, which are key to fluoroquinolone resistance. Please include this analysis if it was performed, or explain why it was not.

Please revise the plasmid replicon nomenclature, including in the tables, to ensure consistency and accuracy. Furthermore, the thresholds of identity and coverage used for plasmid and resistance gene detection must be clearly stated. If default parameters were used, please specify this in the methods section. Science should be reproducible. 

The paragraph and figure discussing virulence factors (Figure 5) add limited value in their current form. Merely reporting the number of virulence genes is not informative. I recommend removing this section. However, if the authors wish to retain it, they should reorganize the figure (e.g., from highest to lowest number of virulence factors) and significantly expand the discussion to highlight the biological relevance of these findings.

Lines 259-272: This section largely repeats the introduction and does not contribute new insights. I recommend removing it or integrating it more effectively into the discussion.

Lines 320-323: This paragraph appears disconnected from the rest of the discussion. Please revise or remove it to improve flow.

Please review the formatting of bla genes. According to current standards, the gene family (e.g., bla) should be italicized, while specific alleles should include subscript formatting.

Although optional, I believe the authors could make better use of the WGS data. For example, MLST information, especially for E. coli, would add considerable value to the manuscript and provide insight into the diversity of the isolates.

Reviewer 2 Report

Comments and Suggestions for Authors

In this study author screened food samples collected from the Apulia and Basilicata regions of Italy. Additionally, antibiotic resistance in the isolated strains was assessed using conventional phenotypic methods, and whole genome sequencing (WGS) was used to predict the presence of resistance genes. In the following section author will find few comments for their consideration. 

Comments:

  1. Page 2 line 63: Please indicate the country of these areas.
  2. Page 3 line 93: Please provide a brief method for library preparation and the software used for the WGS analysis.
  3. Page 4 line 119-124: Consider rewriting this section as from these sentences it looks like you never cultured few samples to isolate ESBL strains.
  4. Page 7 line 15: how the intrinsic resistance was determined?
  5. Page 11 line 245: Author correlated the resistance genes with the MICs so I think it would be nice to see the correlation of virulence gene identified by the bioinformatics analysis to their expression using real time PCR to understand the expression in different samples like RTE or raw samples where they were isolated from. Author can select the common genes identified in all isolates like ompA to perform the real time PCR.

Round 2

Reviewer 1 Report

Comments and Suggestions for Authors

Congrats on the improvements made, it is indeed much better now. However, despite these revisions, I am still conviced the paper does not represent a substancial contribution to the field. That said, I eill leve the final decision regarding acceptance to the editor

Author Response

Comment: Congrats on the improvements made, it is indeed much better now. However, despite these revisions, I am still conviced the paper does not represent a substancial contribution to the field. That said, I eill leve the final decision regarding acceptance to the editor

Answer: We sincerely thank the reviewer for acknowledging the improvements made in the revised version of the manuscript. We also appreciate the time and effort dedicated to the review process. While we regret that the reviewer remains unconvinced of the substantial contribution of our work, we respectfully believe that the manuscript offers an important contribution to knowledge on the distribution of ESBL producing Enterobacteriaceae in food. We have made every effort to clarify the relevance and originality of our contribution in the revised version. We leave the final decision in the capable hands of the editor and are grateful for the opportunity to submit our work for consideration.